# Chemistry and Art of Developing Lipid Nanoparticles for Biologics Delivery: Focus on Development and Scale-Up

**DOI:** 10.3390/pharmaceutics16010131

**Published:** 2024-01-19

**Authors:** Rijo John, Jasmin Monpara, Shankar Swaminathan, Rahul Kalhapure

**Affiliations:** 1Department of Pharmaceutical Sciences, Philadelphia College of Pharmacy, Saint Joseph’s University, Philadelphia, PA 19104, USA; rjohn@sju.edu (R.J.); jmonpara@sju.edu (J.M.); 2Drug Product Development, Astellas Institute of Regenerative Medicine, Westborough, MA 01581, USA; shankarswamin55@gmail.com; 3Discipline of Pharmaceutical Sciences, School of Health Sciences, University of KwaZulu-Natal, Private Bag X54001, Durban 4000, South Africa; 4Odin Pharmaceuticals LLC, 300 Franklin Square Dr, Somerset, NJ 08873, USA

**Keywords:** lipid nanoparticles, biologics, cationic lipid, liposome, nucleic acids, drug delivery systems, phospholipid, cholesterol, ionizable lipid, PEG–lipid

## Abstract

Lipid nanoparticles (LNPs) have gained prominence as primary carriers for delivering a diverse array of therapeutic agents. Biological products have achieved a solid presence in clinical settings, and the anticipation of creating novel variants is increasing. These products predominantly encompass therapeutic proteins, nucleic acids and messenger RNA. The advancement of efficient LNP-based delivery systems for biologics that can overcome their limitations remains a highly favorable formulation strategy. Moreover, given their small size, biocompatibility, and biodegradation, LNPs can proficiently transport therapeutic moiety into the cells without significant toxicity and adverse reactions. This is especially crucial for the existing and upcoming biopharmaceuticals since large molecules as a group present several challenges that can be overcome by LNPs. This review describes the LNP technology for the delivery of biologics and summarizes the developments in the chemistry, manufacturing, and characterization of lipids used in the development of LNPs for biologics. Finally, we present a perspective on the potential opportunities and the current challenges pertaining to LNP technology.

## 1. Introduction

Lipid nanoparticles (LNPs) have emerged as a promising vehicle for various drugs and gene delivery applications since the development of phospholipid vesicles named “liposomes” in the 1960s [1]. The use of LNPs has played a significant role in the development of various immunotherapies and vaccines, such as cancer immunotherapy, anticancer vaccines, and COVID-19 vaccines, as well as in the investigational vaccines for a diverse range of infectious, viral and bacterial diseases currently in preclinical and clinical studies [2,3]. Over the past few years, LNPs have been explored as a potential solution for drug delivery for small- and large-molecular-weight pharmaceuticals, including biologicals. Given the unique structural and biological properties of nucleic acids, peptides, and oligonucleotides, they have been categorized as biologicals for this review. The focus has been on addressing issues and improving the properties of therapeutic moieties by protecting them from in vivo degradation, enabling controlled release, modifying distribution, enhancing targeted delivery, improving solubility, and increasing bioavailability [4,5]. Additionally, nanoparticles can be developed to improve their ability to reach extracellular or intracellular specific targets and to compete with the virus’s ability to adhere to cell surface receptors controlling viral infections, overcoming drug resistance and the passive/active targeting of drugs [6]. Furthermore, these lipid-based carriers can transport hydrophobic and hydrophilic drugs to improve their therapeutic effects [7]. In general, LNPs consist of four primary constituents (Figure 1): cationic or ionizable lipids that bind with negatively charged biological material and assist in endosomal escape, phospholipids that provide structure to the particle, cholesterol that contributes to the stability of the nanoparticles and enables membrane fusion, and pegylated lipids that improve stability and circulation of the nanoparticles [8,9,10].

The development of the first USFDA-approved small-molecule liposomal preparation, Doxil^®^, in 1995, was a landmark in the research on lipid-based nanocarriers and it is rapidly advancing in the areas of cancer treatment and biologics delivery [11]. The encapsulation of doxorubicin within PEGylated liposomes significantly reduced side effects commonly associated with free doxorubicin, such as chronic cardiomyopathy and congestive heart failure [12]. Liposomes have been widely used and have shown significant progress in clinical applications for nearly 30 years. These include products that encapsulate a variety of small drug molecules. Some of the commercial lipid-based drug products available in the market are AmBisome^®^ (Amphotericin-B), DaunoXome^®^ (Daunorubicin), DepoCyt^®^ (Cytarabine), DepoDur^®^ (Morphine), Doxil^®^ (Doxorubicin), Inflexal^®^ V (trivalent influenza vaccine), Mepact^®^ (Mifamurtide), Myocet^®^ (Doxorubicin), Visudyne^®^ (Verteporfin), Amphotec^®^ (Amphotericin-B), Abelcet^®^ (Amphotericin-B), Diprivan^®^ (Propofol), Fungizone^®^ (Amphotericin-B), Estrasorb^®^ (Estradiol), Exparel^®^ (Bupivacaine), Lipodox^®^ (Doxorubicin), Onivyde^®^ (Irinotecan), Nocita^®^ (Bupivacaine), Vyxeos^®^ (Daunorubicin and Cytarabine), Shingrix^®^ (Zoster Vaccine), Lipoplatin^TM^ (Cisplatin), and Arikayce^®^ (Amikacin) [13,14]. The applications of lipids or LNPs are not only limited to small molecular drugs but also include biomacromolecules such as messenger RNA (mRNA), small interfering RNA (siRNA) and antibodies [7,15]. This list can be updated further with the development of nucleoside-based nanomedicines for the treatment of hereditary transthyretin-mediated amyloidosis (Onpattro^®^) and prophylaxis of severe acute respiratory syndrome coronavirus 2 (SARS-CoV-2) infection (Comirnaty^®^ developed by Pfizer/BioNTech Pfizer, New York, NY, USA-BioNTech, Mainz, Germany and spikevax^TM^ developed by Moderna Cambridge, MA, USA, both are injectable suspensions for intramuscular use). The unique properties of lipids in drug delivery systems also pose challenges in drug product development and biopharmaceuticals related to product quality and performance. The successful formulations require a comprehensive knowledge of the chemistry and physical characteristics of lipids. The major factors for the selection of lipids include water miscibility, solvent capacity, purity, chemical stability, digestibility, the fate of digested products, safety, and regulatory profile [15]. LNPs, based on their fabrication methods and physicochemical properties of formulations are classified into various classes such as liposomes, niosomes, and transferosomes, which form the vesicular systems. Solid lipid nanoparticles (SLN), nanostructured lipid carriers (NLC) and lipid polymer hybrid nanoparticles (LPHNs) constitute the particulate systems [16]. Liposomes exhibit remarkable versatility as a nanocarrier platform due to their ability to transport hydrophobic or hydrophilic molecules and hold the distinction of being the first nanomedicine delivery platform to have made a successful transition from concept to clinical application. In liposomes, efficient encapsulation of anionic molecules such as siRNA and miRNA has been achieved by using cationic lipids. These cationic liposomes have been widely employed for delivering large molecules into cells, and they serve as the basis for several commercially available marketed products [17]. Although liposomes have reduced toxicity and the improved targeting abilities of actives in systemic circulation, their dermal application is restricted due to inadequate penetration ability in the stratum corneum. Additional drawbacks include the inadequate encapsulation of hydrophilic drugs and limited storage capacity caused by drug leakage in the medium [18].

Niosomes, formed using nonionic surfactants and cholesterol in an aqueous environment, offer more stable vesicles and longer shelf life compared to liposomes. Although niosomes possess advantageous properties, such as drug delivery capabilities, they are also prone to drug leakage and particle aggregation due to the absence of ionic repulsion, which has been a significant barrier in obtaining USFDA approvals [19]. Transferosomes are composed of phospholipids, cholesterol, and edge activators (EA) and are characterized by their ability to be elastic and deformed as nanoparticles. Incorporating the EA improves their flexibility, resulting in increased tissue permeability. Transferosomes exhibit improved penetration capability and effective entrapment efficiency for lipophilic molecules [20]. However, their oxidative degradation and the high cost of materials remain challenging factors in their manufacturing and scale-up. Advanced generations of LNPs comprising solid lipid nanoparticles, nanostructured lipid carriers, and lipid polymer hybrid nanoparticles showcase more intricate internal structures and increased physical stability [21]. Figure 2 represents the different types of LNPs with characteristics. SLNs consist of a solid lipid core, which is typically glyceride-based, and a surfactant coating that serves to stabilize them. The stability of nanoparticles in vivo is enhanced by the solid core of SLN that remains solid at physiological temperature [22]. They exhibit a higher encapsulation efficiency for hydrophobic drugs because of the absence of an aqueous core. SLNs are associated with low drug loadings, and there is also a possibility of drug expulsion during storage because of the crystallization of solid lipids. Nonetheless, SLNs’ ability for large-scale production and consistent reproducibility is favoring their downstream and clinical applications [23]. NLCs have been developed as a continuation of SLNs to improve the encapsulation efficiency and prevent drug leakage. NLC matrices form amorphous structures that are composed of solid and liquid lipid blends [24]. The larger inter-fatty acid chain distances resulting from the presence of distinct lipid molecules in NLCs create more imperfections (irregularities) for drug accommodation and thus better stability during storage [5]. The higher solubility of drugs in liquid lipids can be used to enhance the drug loading [25]. The development of lipid polymer hybrid nanoparticles represents a new generation of delivery systems that harness the unique characteristics of polymeric nanoparticles and liposomes that were crucial to their early therapeutic success. Additionally, these lipid nanoparticles were designed to overcome drawbacks, such as structural disintegration, a shortened circulation period, and material leakage, that were identified in prior delivery systems [26]. The polymer controls the release of the drug, whereas the lipid enhances drug permeation across the membrane and facilitates loading. Thus, these hybrid nanoparticles made of polymer and lipid can enhance both the physical stability and biocompatibility of drugs, making them a promising option for an effective drug delivery system [27].

The administration of biologic drugs remains a significant challenge because these large molecules often do not conform to Lipinski’s “rule of five” criteria and are prone to be unstable both in vitro due to aggregation and in vivo due to enzymatic and chemical degradation. Delivering biologics to their intracellular target while avoiding intracellular degradation pathways poses a significant challenge for many of these molecules. The delivery of oligonucleotide has been most effective using LNPs as a carrier [28]. The LNPs have been extensively characterized [29] and Pieter Cullis’s [30] research group has reported the use of LNPs as a means of delivering SiRNA, resulting in the product Onpattro^®^ (developed with Alnylam Pharmaceuticals Inc., Cambridge, MA, USA), approved by USFDA in August 2018. This review describes the LNP technology for the delivery of biologics, and summarizes the developments leading to the chemistry, manufacturing and characterization of lipids used in development of LNPs for biologics with clear clinical potential.

## 2. Methods of Manufacture of LNPs

The methods used for formulating LNPs vary from low-energy methods that utilize the thermodynamic, solubility, temperature-dependent properties of the excipients, and high-energy methods that involve high pressure, shear and/or ultrasonic energy input to form nanoparticles from an initially coarsely dispersed mixture of the formulation ingredients. Briefly, the methods of preparation can be classified into (i) high energy (ii) low energy and (iii) organic solvent-based methods. Figure 3, Figure 4 and Figure 5 represents comparison of various techniques used in the production of LNPs, with advantages/disadvantages [31,32].

### 2.1. High-Energy Methods

#### 2.1.1. High-Pressure Homogenization (HPH) Technique

HPH is renowned for its reliability and well-established technique to facilitate large-scale production [33,34]. This method involves using high-pressure homogenizers to force a liquid through a micron-sized gap under high pressure (1 × 10^7^–2 × 10^8^ Pascal). The high pressure causes the liquid to accelerate to a velocity exceeding 277.778 m/s, which generates shear stress and cavitation forces that break down the particles [34]. The quality attributes of nanoparticles is influenced by several factors such as the duration of homogenization, temperature controls, homogenization speed, concentration and the type of surfactant, lipid and drug used. Among these, the drug loading capacity is considerably influenced by Drug’s solubility in lipids. The process can be carried out either at a high temperature (hot homogenization) or at below room temperature (cold homogenization) [35].

During the hot homogenization method, the process is performed at temperatures higher (278.15–283.15 kelvin) than the melting point of the lipid. The active ingredient, molten lipid, and the emulsifiers aqueous phase are combined using a high-speed stirrer to form a pre-emulsion. After homogenizing the resulting oil-dispersed-in-water (o/w) emulsion, it is subsequently cooled to room temperature that generates the LNPs. An advantage of this technique is that it allows for the incorporation of lipophilic drugs [36]. Although these offer numerous benefits, it is an energy-intensive process that can expose heat-sensitive compounds to excessive temperatures.

The cold homogenization process involves the incorporation of an active moiety in the molten lipid, which is cooled immediately using liquid nitrogen or dry ice. These can address the shortcomings of hot homogenization, which may result in the separation of lipophilic substances into an aqueous phase, drug degradation, and complexity of crystallization leading to the modification of nanoparticles. Efficient pulverization of lipids is promoted by the increased brittleness resulting from the low temperatures utilized during the process [36].

#### 2.1.2. Supercritical Fluid Technology (SCF)

SCF technology offers a promising approach for producing nanoparticles, providing several benefits such as precise control over particle size, consistent size distribution and the removal of solvents [37,38]. SCF technology takes a supercritical form and can have its solvent power modified by changing the pressure and temperature [39]. In brief, supercritical carbon dioxide (scCO_2_) acts as a solvent and the solubilities of solid lipids and drugs are enhanced in scCO_2_ upon being pumped into the high-pressure vessel. Subsequently, the solid lipids and drugs are subjected to a depressurization process, causing them to become supersaturated and precipitated out, which forms drug loaded LNPs [40]. SCF technology offers a significant advantage in the preparation of LNPs by enabling a substantial increase in the solubility of a substance in the liquid through small pressure changes. In a typical process, controlled depressurization of an SCF can promptly generate the supersaturation of dissolved compounds, leading to their precipitation, although SCF technologies provide advantages in the production of LNPs, the high cost of SCF equipment could be a limiting factor. However, utilizing precise computational modeling of the SCF manufacturing process could be a valuable tool for enhancing these technologies [41].

#### 2.1.3. Ultrasonication

The principle behind this method involves the utilization of sound waves to decrease the size of particles. This method is dispersion-based and includes melting the lipid matrix along with the drug at a temperature of 278.15–283.15 kelvin above its melting point. After melting, the lipid is dispersed in an aqueous phase that contains a surfactant, with rapid stirring to generate an emulsion. Ultrasound is applied to the entire mixture to minimize the size of droplets and the mixture is subsequently cooled gradually to yield the nanoparticle dispersion. The use of commonly available laboratory equipment is the most prominent advantage of this method, for small-scale production. However, the presence of formulations with a high degree of polydispersity and numerous microparticles can pose challenges for drug delivery approaches based on LNPs [42,43]. This technique is widely used in the preparation of SLNs on a laboratory scale.

#### 2.1.4. Flash Nanocomplexation (FNC)

Plasmid DNA/polycation nanoparticles have been successfully synthesized using this method, which relies on the complexation of polyelectrolytes in aqueous solutions, protein-loaded chitosan nanoparticles, and protein antigen/oligonucleotide adjuvant co-encapsulated nano-vaccines. The introduction of multiple jets inside the chamber of a multi-inlet vortex mixer (MIVM) generates turbulence, facilitating instantaneous and highly efficient mixing at a molecular level [44]. By maintaining a homogenous mixing condition throughout the mixing chamber, nanoparticle assembly can take place, leading to the generation of uniform nanoparticles without the necessity of post-production processing [45,46]. The optimization of nanoparticle characteristics, such as size and shape, and their impact on nanoparticle transport kinetics, can be achieved by adjusting factors like jet velocity, the concentrations of various components, and the complexation capacity of each polyelectrolyte component. LNPs generated through FNC utilize a TFF (Tangential Flow Filtration) process to purify and concentrate all the LNPs. KrosFlo^®^ Research 2i TFF system (Repligen, Waltham, MA, USA) equipped with a 100 kDa mPES filter (ID 0.5 mm). TFF utilizes diafiltration, a process that involves the selective removal of permeable molecules from a solution as it flows through a hollow ultrafiltration membrane. The application of diafiltration for nanoparticle purification in colloidal dispersions has been well characterized [47]. TFF provides a pressure-driven one-way permeation that enables the purification of a wide range of liposome nanoformulations, achieving high lipid recovery rates of over 98%. It efficiently eliminates non-encapsulated compounds (>95%) and organic solvents (reduction of >95%) within a short period of time. Filtration is performed by directing the media parallel to and through the membrane, effectively preventing the accumulation of molecules on the membrane. This process continues until the entire volume has fluxed.

### 2.2. Low-Energy Methods

#### 2.2.1. Thin-Film Hydration Method

The thin-film hydration method uses organic solvents (dichloromethane, ethanol, chloroform and a methanol–chloroform mixture) to dissolve lipids; subsequently, the organic solvent can be removed through evaporation under vacuum at a temperature of 318.15–333.15 kelvin, resulting in the formation of a thin lipid film. Further, the thin lipid film gets hydrated in aqueous media by continuous agitation for 2 h at a temperature of 333.15–343.15 kelvin, where it swells to produce liposomes with an aggregation of multilamellar vesicles. These multilamellar vesicles can be downsized either by extrusion through a polycarbonate membrane (French pressure cell), using probe sonication, or through a microfluidizer to attain size uniformity, lamellarity, and nanoparticle distributions. The method of incorporating drugs into liposomes depends on the characteristics of the drug. If the drug is lipophilic, it can be dissolved with the lipids during the liposome preparation process. On the other hand, hydrophilic drugs can be incorporated during hydration or by active loading into the liposomes [48].

#### 2.2.2. Reverse Phase Evaporation Method

The lipids are dissolved in a low-boiling-point organic solvent and subsequently mixed directly with water or a buffer that contains a water-soluble drug. By utilizing a low-pressure rotary evaporator, the organic solvent is evaporated, and the proportions of both phases are reversed, which generates lipid nano-liposomes dispersed within the aqueous phase. Extrusion or sonication methods can be employed to achieve a reduced particle size and promote mono- and polydispersity in preformed LUVs and MLVs [49].

#### 2.2.3. Detergent Removal Method

The detergent removal method involves hydrating and solubilizing lipids by utilizing a solution of detergents [50]. Mixing causes the detergent to bind with the phospholipids, resulting in the formation of mixed micelles comprising both detergent and lipids. As the detergent is removed in a successive or progressive manner, the mixed micelles become better off in lipids, resulting in the generation of unilamellar vesicles. Sodium cholate, Triton X-100, sodium deoxycholate, and alkyl glycoside are examples of detergents commonly utilized due to their high critical micelle concentration (CMC). The surfactant can be removed through various methods such as a dialysis membrane, size-exclusion chromatography, or adsorption onto lipophilic beads [51].

#### 2.2.4. Dehydration–Rehydration Method

In this technique, the lipids are dispersed in small quantities into an aqueous phase containing the drug, which is then followed by sonication. Initially, the water molecules are vaporized through a dehydration process using nitrogen, resulting in the formation of a multilayered film that encapsulates the drugs. Subsequently, a hydration step is conducted to create large unilamellar vesicles (LUVs) that contain the drugs [52].

#### 2.2.5. Microfluidic-Assisted Method

The nanoparticles are formed by a controlled transition from jet to drip, which is achieved with a specific chip geometry when liquid droplets are propelled into a carrier fluid. To prevent the coagulation and separation of NPs, surfactants are employed as stabilizing agents. Microfluidics offers the possibility of the on-chip preparation of liposomes and other lipid-based nanoparticles, allowing for precise control over the mixing of aqueous and organic phases. As a result, liposomes with improved characteristics can be consistently achieved in a more reproducible manner [53].

#### 2.2.6. Microemulsion

Microemulsions are pseudo-ternary systems composed of oil, water, and surfactant, frequently used in combination with cosurfactants that display distinct physicochemical attributes such as clarity, thermodynamic stability, low viscosity, and an isotropic nature. These microemulsions, characterized by stable single-phase swollen micelles, form spontaneously and possess the advantage of incorporating significant quantities of both lipophilic and hydrophilic drugs [54].

#### 2.2.7. Double Emulsion

The main purpose of this method is to prepare LNPs that are loaded with hydrophilic drugs as well as various biological molecules such as peptides and insulin [55]. In this technique, the drug is first dissolved in an aqueous phase and then emulsified in the molten lipid along with an emulsifier to stabilize the primary emulsion. This primary emulsion is dispersed into an aqueous phase that consists of a hydrophilic emulsifier. Further, the double emulsion is mixed and subsequently separated by sifting. Physical instability such as particle growth during storage and broad particle size distribution are the significant drawbacks of this technique [56].

#### 2.2.8. Phase Inversion Technique

The phase inversion method involves mixing of different formulation components like drugs, a lipid matrix, water, and surfactant on a magnetic stirrer under three temperature cycles (333.15–358.15 kelvin). Following this, the mixture is subjected to a sudden temperature change by diluting with cold-distilled water, resulting in the formation of LNPs. This technique offers the benefit of not requiring organic solvents and less exposure to heat. Nonetheless, it can be a time-consuming process. The stability of LNPs following their manufacture relies on the storage temperature relative to their phase inversion temperature and melting/crystallization points [57,58].

### 2.3. Organic Solvent-Based Methods

#### 2.3.1. Solvent Emulsification—Evaporation Technique

In this method, the drug and lipids are solubilized in non-polar organic solvents such as toluene, cyclohexane, dichloromethane, or chloroform. Emulsification is then performed in an aqueous phase with the use of a high-speed homogenizer. Finally, the organic solvent is removed through stirring at room temperature (298.15 kelvin) under reduced pressure (4000–6000 pascal). Mechanical mixing at a reduced pressure and temperature causes the precipitation of lipids, resulting in the generation of lipid nanoparticles [59]. This method is appropriate for encapsulating heat-labile drugs. Nevertheless, it may be challenging to completely remove organic solvents, especially if the lipids are not very soluble in the solvent. This can potentially lead to toxicity by residual solvents.

#### 2.3.2. Solvent Emulsification—Diffusion Technique

This approach involves emulsifying the lipid matrix in water under reduced pressure. The lipids are precipitated in an aqueous environment, resulting in a dispersion of nanoparticles with an average diameter ranging from 30 to 100 nm. The solvent of choice such as butyl acetate, ethyl acetate, benzyl alcohol, isopropyl acetate, or methyl acetate must have partial miscibility with water. Since this method does not necessitate the use of heat, it is suitable for thermolabile drugs [60,61].

#### 2.3.3. Solvent Injection

This technique involves dissolving the lipid matrix in a water-miscible solvent and rapidly injecting the resulting mixture through a needle into a mixed aqueous phase that includes the surfactant, or without surfactant under constant stirring, and the dispersion is filtered. In this method, the process parameters for synthesizing nanoparticles encompass the properties of the injected solvent, the concentration of the lipid, the amount of the injected lipid solution, and the viscosity, as well as the diffusion of the lipid solvent phase into the aqueous phase. This technique offers several advantages, including simple handling procedures and a rapid production process that does not necessitate the use of sophisticated equipment. However, the utilization of organic solvents is a drawback of this method [62]

#### 2.3.4. Membrane Contactor Technique

This approach utilizes a modified ethanol injection technique, which involves a pair of pressurized vessels. One vessel holds an organic phase containing lipids, while the other contains an aqueous phase. These two phases are separated by a unique porous glass membrane with specific pore sizes that enable the flow of the organic phase. Polypropylene hollow fibers, often used as the membrane, are preferred due to their capability to accommodate larger surface areas and facilitate uniform fluid flows [63]. The lipid phase is forced through membrane pores at a temperature above the lipid’s melting point, resulting in the formation of small droplets. As the aqueous phase circulates within the membrane module, it sweeps away the droplets that are formed at the pore outlets, and lipid nanoparticles are formed by cooling the product at room temperature [64]. The process parameters that influence the generation of lipid nanoparticles include the lipid-phase and aqueous-phase temperature, membrane pore size, lipid-phase pressure, and aqueous-phase crossflow velocity. By maintaining the aqueous phase temperature below the lipid’s melting point, the solidification of the lipidic phase in the aqueous phase is instantaneous, leading to the generation of nanoparticles. This method is simple with no significant drawbacks and the size of the nanoparticles can be controlled by utilizing the membranes with varying pore sizes.

### 2.4. Large-Scale Production of LNPs

The large-scale production approaches for development of LNPs are essential to establish nanoparticles potential in pharmaceutical applications. High-pressure homogenization is a proven method for scaling up nanoparticle production and has been used since 1950s, when it was used to create parenteral emulsion [65]. Pilot-scale studies have demonstrated that HPH can produce both drug-free and drug-loaded LNPs. A systematic analysis was reported for creating Stavudine-loaded LNPs through the HPH method, starting from a laboratory scale and progressing up to an industrial scale [66]. HPH is a favored technique for producing LNPs due to its ease of scale-up, absence of organic solvents, and shorter production times, which make it a practical and environmentally friendly option for industrial applications. LNP production techniques, including HPH, are often hindered by several common challenges such as drug degradation during the manufacturing process, lipid crystallization, gelation phenomena, supercooled melts and changes in lipid and particle shape. However, it is possible to manage these limitations by carefully analyzing production conditions such as temperature range, shear stress, and light, and by improving the selection of drug carriers, formulation, and drug loading methods [67]. Hot melt extrusion coupled with HPH has the potential to create a scalable process for producing LNPs [68]. This involves feeding the raw materials into the extruder barrel at a temperature higher than the melting point of the lipids used and further reducing the size of the LNPs by attaching a high-pressure homogenizer to the end of the hot-melt extruder barrel using an insulated connector. The concentration of lipids, screw design, and residence time are the process parameters that have the greatest influence on the size of LNPs among all the parameters studied. Gasco et al. utilized the microemulsion method to fabricate LNPs [45]. In industrial applications, it is common to produce the microemulsion within a temperature-regulated tank and then transfer it to a tank of cold water for precipitation. Both methods mentioned above necessitate a heating process that could result in the degradation of heat-labile drugs. Additionally, the HPH method demands an intensive amount of energy, while the microemulsion method employs surfactants. Bulk nanoprecipitation is a suitable method for producing LNPs on a small scale, but achieving perfect mixing with a short mixing time becomes challenging when scaling up to a larger volume. Hence, to enable large-scale production, various mixing devices such as confined impinging jet mixers, microfluidics, and T-mixers have been developed. The synthesis of LNPs through continuous nanoprecipitation has been made possible with the development of microchannel mixers, but the productivity of the process remains low. Microchannels are channels that contain a cross-junction for both lipids and aqueous solutions, along with a T-shaped junction for the insertion of gas [69]. The formation of a lipid solution using a surfactant and water-based organic solvent simultaneously along the cross-junction into the mainstream is employed to focus flow of liquid. To achieve gas displacement, an inert gas is injected into the microchannel’s upward main flow, creating a slug flow of the gas–liquid mixture through the T-shaped junction. This liquid flow-focusing technique, based on hydrodynamic focusing, can produce LNPs with small diameters and narrow size distributions. Static mixers consisting of numerous identical and static elements with intricate structures such as tubes, columns, or reactors have been created for the continuous and large-scale production of LNPs [70]. The size of LNPs was found to be significantly affected by the lipid concentration, with an increase in the lipid concentration leading to an increase in particle size.

### 2.5. Post-Production Processing for LNPs

#### 2.5.1. Sterilization

Autoclaving (steam sterilization), gamma irradiation and filtration are the techniques employed for sterilizing LNPs. Radiation sterilization is considered a recognized method by regulatory agencies to meet the sterility criteria of parenteral products, especially thermosensitive drugs. However, radiation can potentially cause the chemical degradation of lipids as well as actives [71]. Steam sterilization is a widely employed method for sterilizing objects that can withstand high temperatures (394.261–413.706 kelvin) (394.15–413.15 kelvin) and pressure (around 110,316–241,317 pascal) [72]. This method is one of the most effective ways to achieve sterilization as it does not affect mean particle size and zeta potential if exposed for short time. Aseptic manufacturing techniques with sterile filtration are typically necessary for biologic drug substances due to their susceptibility to degradation when exposed to heat, radiation, or chemicals. Sterile filtration techniques have found extensive application in the manufacturing of biotherapeutics, including monoclonal antibodies (mAbs) and recombinant DNA-derived proteins. In the production of biotherapeutics, the application of sterile filtration is essential during the preparation of buffer and cell culture media.

Sterile filters function through the process of normal flow filtration, whereby the membrane effectively retains bacteria, cell debris, and insoluble aggregates. These filters are employed for the removal of bacteria and particles from feedstock solutions, safeguarding downstream units against fouling caused by insoluble materials, and enabling sterile fill operations. Sterile filters typically include filters with pore sizes of 0.1 and 0.2 μm, both meeting precise standards for effectively removing microorganisms. The criteria for sterile filtration using 0.2 μm filters rely on the elimination of 10^7^ colony-forming units (CFU) of *Brevimunda diminuta* per square centimeter of membrane surface. Polyethersulfone (PES), polyvinylidene fluoride, nylon, and polypropylene (PP) are among the diverse base polymers used to produce sterile filtration membranes [73]. The sterilization of LNPs presents a major challenge due to the risk of destabilization from conventional methods. For instance, the common use of γ radiation for sterilization can result in lipid oxidation and chain fragmentation, impacting both the stability and efficacy of LNPs. Autoclaving has the potential to initiate phase transitions and thermal stress, resulting in the further destabilization of LNPs. The use of filtration and aseptic processing can induce aggregation or deformation owing to shear stress. In response, various alternative sterilization methods such as UV irradiation, ethylene oxide sterilization, and gentle sterile filtration have been investigated as potential solutions. Therefore, the importance of choosing an adequate sterilization method persists to uphold the stability and effectiveness of LNPs throughout industrial production [34].

#### 2.5.2. Lyophilization

The lyophilization process of the synthesized LNPs is crucial as it ensures chemical and physical stability in long-term storage, which is a critical factor especially for products containing hydrolyzable drugs. The process of lyophilization involves freezing the material, followed by the application of reduced pressure and heat to enable the frozen water within the material to sublimate. Lyophilization conditions, although effective in removing water, can promote the aggregation of lipid nanoparticles, but the use of an appropriate amount of lyoprotectants can minimize this effect [74].

## 3. Characterization Techniques of LNPs

The characterization of LNPs is essential for investigating the synthesis of nanoparticles and also for controlling quality to meet the requirements of pharmaceutical applications in the successful development of drug delivery systems. The uptake and distribution of LNPs are significantly influenced by physicochemical factors such as size, surface charge, molecular weight, and solubility. Hence the significant properties of LNPs that need to be characterized include the particle size, polydispersity index, zeta potential, surface morphology, entrapment efficiency, crystallinity, drug release and stability [75].

### 3.1. Particle Size, Polydispersity Index (PDI), Zeta Potential (ZP) and Surface Morphology

The size of particles plays a crucial role in nanoparticle drug delivery applications. LNPs typically have an average particle size ranging from 100 nm to 400 nm. However, to achieve optimal systematic drug delivery through intravenous (IV) injection, LNPs with particle sizes ranging from 10 nm to approximately 150 nm are preferred [76]. The polydispersity index (PDI) provides information about the range of particle sizes, with a scale ranging from 0 to 1. A PDI value of less than 0.2 is typically considered indicative of a narrow size distribution, while many studies set the upper limit for the PDI value at less than 0.3. PDI values below 0.05 are typically observed in samples with highly uniform particle sizes. Conversely, PDI values greater than 0.7 are indicative of a broad distribution of particle sizes. Dynamic light scattering (DLS) and quasi-elastic light scattering are techniques that can be used to measure particle size, the polydispersity index, and charge analysis [77]. DLS, used to determine both the particle size and PDI [78], works by measuring the variations in the intensity of scattered light resulting from the Brownian motion of particles. The extent of charge on the surface of particles in an aqueous dispersion is specified by the ZP, which is a critical parameter in predicting the long-term physical stability of formulations [79]. A ZP value of LNPs is typically estimated using electrophoretic mobility and measured using techniques such as photon correlation spectroscopy [80]. The occurrence of particle aggregation is prevented by a high repulsion force. In general, particles are deemed electrostatically stable and capable of repelling each other when their ZP is above +30 mV or below −30 mV. The addition of nonionic steric stabilizers, such as polyhydroxy surfactants, to formulations is expected to result in a decrease in ZP value. However, in SLNs and NLCs, the ZP value has been found to increase as the oil content in the formulations increases [81]. Transmission electron microscopy (TEM), scanning electron microscopy (SEM) and atomic force microscopy (AFM) methods can be utilized for the direct imaging of the shape and the dimensional analysis of the size of LNPs [37]. SEM and TEM use electrons that are transmitted from the surface and inner structure of particles, respectively, to provide morphological information. In contrast, AFM can generate a three-dimensional profile of LNPs. AFM is a significant tool as it enables the access of atomic-level resolution, in addition to size, colloidal attraction, and deformation resistance. AFM has been utilized to investigate the morphology and surface of LNPs at the single-particle level. Further, it was employed to study the alterations made to transferrin, focusing on the development of small spherical structures on the surface of liposomes. Additionally, confirmation of dimeric and trimeric transferrin assemblies was obtained by sizing these nanostructures. Besides analyzing size and morphology, AFM-based nanoindentation offers a new perspective on the mechanical attributes of LNPs, which are important in the cellular uptake process mediated by endocytosis [82]. AFM is a relatively simple method in visualizing nanoparticles in a liquid environment with minimal disturbance within the pico-to-micro-Newton range; however, technical difficulty in immobilizing particles onto a solid substrate is a challenge. Yuki Takechi-Haraya et al. devised a method employing AFM to observe mRNA-LNPs within an aqueous solution, assessing their size and structure. This approach involves utilizing an anti-PEG antibody targeted at the PEG chain of mRNA-LNPs, facilitating the particles’ immobilization on a substrate [83].

### 3.2. Crystallinity

The analysis of lipid crystallinity or polymorphic modifications can be performed using differential scanning calorimetry (DSC) and X-ray diffraction techniques (XRD) [84]. Assessment of crystallinity in LNP components holds significant importance as the lipids and drugs enclosed within may undergo polymorphic transformations while in storage. This could ultimately lead to instability and drug expulsion, highlighting the need to determine crystallinity levels [85]. Crystallinity would also affect drug entrapment and drug release. DSC is an analytical method that utilizes heat to determine the nature of lipid particles and accurately establish their degree of crystallinity by measuring the enthalpy and peaks observed at different phase transition temperatures. The measurement of enthalpy change with DSC provides a simple and rapid approach for determining the degree of crystallinity in LNPs. However, one significant limitation of this technique is that it is destructive, potentially altering or destroying the sample being analyzed. Powder XRD is a non-destructive technique utilized to analyze the crystal structure of LNPs and determine the crystallinity of the lipid [86]. The crystallographic structure of the tested samples can be determined using XRD, which measures the intensity and angle of X-ray scattering through the samples. Nevertheless, XRD has limitations as it can only analyze powder samples, requiring the LNP-containing suspension to undergo a drying process that may lead to polymorphic transitions. To minimize drug leakage during polymorphic transitions and achieve a low degree of crystallinity, it is recommended to employ both solid and liquid lipids in the elaboration of the lipid matrix. This approach creates additional space for accommodating therapeutic moieties [87].

### 3.3. Entrapment Efficiency (EE)

The drug entrapment efficiency (EE) serves as an important indicator in the assessment of the drug encapsulation preparation method.
$$\%EE=\frac{Encapsulated\;drug\;amount\;by\;nanoparticles}{Total\;drug\;amount\;}\times 100\%$$

Analytical techniques such as UV spectrophotometry or high-performance liquid chromatography (HPLC) and separation techniques (including ultrafiltration, centrifugation, and dialysis) are used to determine the entrapment efficiency. The two primary approaches for measuring the entrapment efficiency are the direct and indirect methods. The direct method involves directly measuring the amount of drug that is encapsulated, while in the case of the indirect method, the quantity of unencapsulated drug in the supernatant is measured [88]. The direct method is typically more suitable for measuring the entrapment efficiency of lipophilic drugs, whereas the latter (indirect method) is more appropriate for hydrophilic drugs. To optimize drug delivery, it is preferable to achieve a higher EE. An EE higher than 70% is the usual expectation for drugs encapsulated in LNPs. Several critical factors, such as the types of lipid materials used, their composition and crystallinity, and the drug’s solubility in both organic and aqueous phases, can impact the EE [88]. The entrapment efficiency of these LNPs is primarily influenced by the type of lipid, concentration, and crystal structure. In addition, the entrapment efficiency can also be impacted by the drug partitioning between the melted lipid and aqueous medium. The decrease in the solubility of a drug in lipids due to the cooling of molten lipids depicts the importance of determining the amount of drug mixed with the lipid particles and the amount of drug solubilized in other structures within the formulation [89].

### 3.4. In Vitro Drug Release Study

Biodegradation and diffusion processes are the primary factors governing the drug release profile from LNPs. Phosphate-buffered saline (PBS) or simulated body fluids are commonly employed as mediums for conducting in vitro drug release studies that aim to simulate in vivo drug release and analyze it using a UV spectrophotometer or HPLC. The method of preparation, drug solubility in the lipid, drug/lipid interactions, type of surfactant, composition of the lipid matrix, degree of crystallinity and particle size are all significant factors that affect drug release from LNPs. The in vitro release profile assists in revealing the kinetic behavior and release mechanism of the drug [90]. LNPs require stability to effectively transport drug molecules to their targets while also possessing the ability to release the drug at the precise intended location. External stimuli can be utilized to assist in drug release. For instance, variations in pH levels enable ionizable lipids to trigger drug release selectively at the intended site. The release of drugs from LNPs that contain cationic lipids can also be activated by exchanging lipids with the biomembrane, inducing a nonlamellar phase formation.

SLN and NLC can achieve a more precise control over the release of their drug payloads due to the limited movement of molecules in the solid state. Further, SLNs face challenges due to their tendency of a high initial drug release and leakage during storage, attributed to solid lipid crystallization after the synthesis cooling process. PEGylation is a surface modification strategy for LNPs that could enhance the release rate by decreasing the surface tension of nanoparticles [91,92]. A substantial challenge in LNPs studies is the potential for burst release with the system. The drug-release profile from LNPs can be modified significantly by changing the lipid matrix, surfactant concentration, and production-related elements (e.g., temperature). Moreover, pivotal factors determining the drug release from lipid nanoparticles include manufacturing parameters like surfactant concentration, temperature, and inherent qualities of the lipid matrix [93].

### 3.5. Stability of LNPs

The progress in establishing stable LNPs is still in its early phases and entails various challenges to produce commercial products that incorporate LNPs. The stability profile of LNPs can be assessed by measuring the particle size, PDI, ZP, %EE and drug release profile over different storage periods as per the ICH guidelines [94]. The experience of formulating therapeutic proteins, such as monoclonal antibodies, can be helpful in predicting potential issues that may occur during the preparation of LNPs for RNA delivery. This knowledge can assist in anticipating challenges related to the chemical stability of the LNP components, the physical stability of the LNP (such as disintegration, aggregation, and adsorption on surfaces), and the stability of RNA within the LNP, which can help us gain a better understanding of LNP preparation and enhance the delivery of RNA therapeutics. SLN stability is primarily determined by the type of lipid material used, the concentration of surfactant, and the optimization of temperature during preparation, making these parameters a critical factor to be considered to ensure proper stability and storage [95]. Synthetic nanoparticle studies conducted in aqueous solutions have revealed that several interactions between nanoparticles and co-solutes are foreseeable. These interactions are affected by the hydrophobicity of the nanoparticles, which, in turn, causes diverse behaviors in the effect of salts on the solubility of the nanoparticles [96]. The significance of lipid-based formulations in solving solubility and enhancing the bioavailability of drugs cannot be overstated. However, there is a noticeable challenge regarding lipid excipient chemistry and its performance, particularly concerning oxidative stability. Antioxidants have been utilized to preserve the stability of pharmaceutical products by either intervening chemically in crucial oxidative phases, preventing the onset of reactions, or by markedly reducing oxidative reaction rates. This maintains the integrity of the dosage form and ensures the stability of the drugs. Chelating agents have the potential to serve as an initial protective measure against oxidation initiated by metals. Optimal antioxidant selection involves assessing solubility within the intended formulation phase, particularly in areas where the oxidation-sensitive substrate (e.g., active pharmaceutical ingredient) is expected to be present [97].

The freeze-drying of LNPs is essential to develop a solid dosage form that can be reconstituted and administered through various routes, such as parenteral, oral, nasal or pulmonary. The freeze-dried cake’s prompt reconstitution upon adding the medium relies on various factors. Undesirable challenges in the freeze-dried powder’s physical structure can result in inadequate wetting, dispersion and capillarity, clumping, undesired particle formation during reconstitution, or an extended reconstitution period. In general terms, the dispersion of liposomes is often linked to drug leakage, recognized as the predominant drawback in the freeze-drying process, especially when dealing with hydrophilic drugs that lack interaction with the bilayer. The main cause of drug leakage is the vesicle fusion or gel to liquid crystalline phase transitions of membrane lipids during the drying process, leading to the subsequent loss of the entrapped material [98].

## 4. Typical Product Attributes of LNPs

The LNPs must contain enough active moieties and may be tailor-made to possess target properties for specific tissues for the site-specific delivery of actives. Structural factors, such as particle size, surface charge, PEGylation, and surface modification with targeting ligands, have been demonstrated to be critical components in regulating the delivery efficacy of lipid-based nanoparticles [2,99]. According to a study conducted by Oussoren et al., the subcutaneous injection of 40 nm liposomes resulted in higher lymphatic uptake compared to larger particles [100]. Liver targeting after systemic administration is only possible through particles with a size smaller than 100 nm, which are capable of diffusing through the liver fenestrae and reaching both hepatocytes and hepatic stellate cells [101]. Large-sized liposomes (>150 nm) were found to be taken up by antigen-presenting cells (APCs) at the injection site and subsequently transported to the lymph nodes. Research has demonstrated that these liposomes exhibit a higher affinity for cells within the lymph nodes [102]. Nanoparticles exhibit an increase in their surface-area-to-volume ratio, reactivity, circulation time, and uptake efficiency as a result of their small size. The conventional liposomal nanoparticles demonstrated an improvement in uptake efficiency with decreasing total size; specifically, liposomes that were 59 nm in diameter exhibited a 2.5-times greater tumor uptake than liposomes that were greater than 100 nm in diameter [103]. Although nanoparticle size is a relevant factor, it is not the sole determinant of cellular uptake. The composition of nanoparticles is a crucial aspect that influences cellular uptake, and it has been demonstrated to exhibit tissue-specific behavior [104].

LNPs predominantly comprise charged molecules, causing an increase in the forces of attraction or repulsion between individual nanoparticles and other components in the surrounding environment. In a colloidal system, the nanoparticles’ behavior is governed by the equilibrium between the repulsive forces they exert on each other, their Brownian motion, and the gravitational force, which collectively maintain their stability [105]. According to Mai et al., there was a significantly higher level of association between anionic and cationic liposomes with B cells compared to uncharged liposomes within the microvascular network. It was observed that the B cell receptor exhibited a greater degree of interaction and internalization with cationic liposomes, while anionic liposomes tended to primarily attach to the surface of B cells [106]. The retention of LNPs at the injection site has been demonstrated to be more significant with cationic particles in comparison to neutral and anionic particles. This effect can be attributed to the strong electrostatic interaction between cationic LNPs and tissues that carry a negative charge. Moreover, cationic LNPs have been found to bind non-specifically with plasma proteins, which has been linked to a higher level of immunogenicity [107]. Positively charged LNPs have a strong affinity for cells but limited efficacy, while negatively charged LNPs are efficiently transported to the lymph nodes. By capitalizing on this charge aspect, charge reversible LNPs have been engineered to optimize gene delivery [108]. The lipid pKa is a crucial determinant of LNP effectiveness, and multiple studies indicate that an optimal pKa of 6.4 is best for achieving the maximum transfection of siRNA-LNPs. However, the optimal range shifts to 6.6–6.8 in the case of mRNA [109]. PEGylation has become a highly sought-after method for lipid-based nanoparticles, as it aims to reduce particle clearance from the bloodstream and subsequently enhance their retention and uptake in targeted tissues or organs. Studies have demonstrated that anionic liposomes modified with PEGs could achieve enhanced clearance at the site of subcutaneous administration and increased retention in the lymph nodes when compared to liposomes that were not modified. Liposomes modified with shorter PEG chains exhibited higher retention but lower clearance in the lymph nodes compared to liposomes modified with longer PEG chains. Therefore, the targeting behavior and transfection capability of LNPs can be significantly influenced by the selection of a linear or branched PEG chain [110]. In addition to passive cellular uptake and prolonged circulation, nanoparticle uptake into target cells can be improved by attaching a receptor ligand to the particle surface. Enhancing the uptake of nanoparticles into target cells can be achieved through methods other than relying on prolonged circulation and passive cellular uptake, such as utilizing a receptor ligand that is conjugated to the particle surface. The incorporation of targeting ligands into lipid-based nanoparticles is a strategy for efficiently targeting delivery systems to specific tissues or cells, including the lymph nodes and other desired locations. The ASSET platform (Anchored Secondary scFv Enabling Targeting) was developed through the conjugation of anti-Ly6C antibodies onto lipid-based nanoparticles that were loaded with siRNA, which selectively targets inflammatory leukocytes in vivo [111,112]. Cell-penetrating peptides have demonstrated their effectiveness in facilitating the intracellular delivery of a diverse range of pharmacologically interesting molecules across various types of cells and show promise in enhancing the intracellular delivery of a wide array of biologically active agents [113]. LNPs have been modified with cell-penetrating peptides, such as R8 and GALA, to improve the uptake of these particles in dendritic cells and to enhance the immune response [114]. The exposure of LNPs to biological fluids causes the biomolecules to be rapidly adsorbed onto their surface, generating the protein corona. Initially, the protein corona comprises significant quantities of low-affinity proteins prevalent in the biological environment. As time progresses, the corona’s makeup becomes enhanced with high-affinity proteins [115].

The presence of a protein corona on LNPs’ surfaces enhances effective cellular uptake, specifically lipoproteins; notably, apolipoprotein E (ApoE) plays a significant role in mediating the binding to LDL receptors situated in plasma membranes, contributing significantly to the process of cellular uptake. The development of a protein corona affects the chemical and physical characteristics of LNPs, increasing the hydrodynamic diameter and modifying the charge on the surface. Nevertheless, all protein coronas contained different patterns of proteins related with the immune response, the metabolism of lipids and the transport capacity, to modulate, in diverse ways, LNP–cell interactions [116]. The initial physicochemical characteristics, including particle size, surface properties, composition, and structure, predominantly determine the biological fate of LNPs within the gastrointestinal tract. Absorption primarily takes place in the small intestine. Notably, LNPs can be categorized as either digestible or indigestible based on their composition, resulting in two distinct biological paths for LNPs. Digestible LNPs undergo digestion initiated by lipases and subsequent hydrolysis across the GI tract upon ingestion. This process leads to the breakdown of lipids, resulting in lipid digestion products rich in fatty acids and monoglycerides. These products, along with encapsulated drugs, bile salts, and phospholipids, form mixed micelles. After absorption by enterocytes, these lipid digestion products and encapsulated drugs enter the systemic circulation through either the portal vein or the lymphatic system. The specific route taken depends on the lipid structure within the LNP formulation and the hydrophobic nature of the encapsulated drugs. Alternatively, indigestible LNPs can remain intact and potentially penetrate the mucus layer to reach epithelial surfaces. After penetration, these LNPs can be transported through the tight junction, or absorbed up by enterocytes or transported by M cells, to enter the circulation [117].

## 5. Chemistry of Lipids Used in the Development of LNPs

Lipids are characterized as amphiphilic molecules due to their structural composition that comprises three domains: a polar head group, a hydrophobic tail region, and a linker that connects the two domains. Cationic lipids, ionizable lipids, zwitterionic lipids and other types of lipids [118] (Table 1) have been explored for their potential in delivering nucleic acids in general, with more emphasis in the past few years on their capability to deliver mRNA. These lipids are composed of a positively charged amine head group and a hydrocarbon chain or cholesterol derivative, linked together through a linker such as glycerol, making them amphiphilic in nature. The head group of the lipid, which carries a positive charge, has the ability to form an electrostatic interaction with nucleic acids that are negatively charged, thereby enabling them to entrap within lipid-based nanoparticles [119,120].

## 6. LNPs in Development/Clinical Trial Pipeline in Industry

LNPs have been identified as the most optimized and effective delivery system for nucleic acids, since negatively charged nucleic acids disrupt their own delivery to cell membranes and are susceptible to degradation by natural enzymes within the body [138]. The completion of the clinical study on LNP-encapsulated siRNA for transthyretin is particularly noteworthy given that this treatment received USFDA approval in 2018. In addition to achieving similar success, there is an ongoing clinical trial for genome-editing therapy. Besides these trials, the majority of the work involves therapies primarily focused on combating cancer. ALN-TTRsc for the treatment of transthyretin in mediated amyloidosis and ALN-PCS02 for treating hypercholesterolemia is currently in clinical trial by Alnylam Pharmaceuticals [139]. The recent report unveiled the outcomes of Phase I in the clinical trial employing CRISPR-Cas9-mediated in vivo gene editing for transthyretin amyloidosis (NTLA-2001). The clinical trial was structured following the preclinical trial conducted in non-human primates, where doses ranged from 1.5 to 6.0 mg/kg, revealing no adverse events within a span of 12 months. This therapeutic CRISPR-Cas9 system identified seven potential target sites [140]. The activation of T cells’ ability to destroy tumors is heavily influenced by the checkpoint inhibition resulting from the interaction between PD-1/PD-L1 and CTLA-4 in solid tumors. There is an active clinical study employing LNP as a carrier for therapeutic mRNA, exploring this specific mechanism [141]. The SNALP technology developed by Tekmira Pharmaceuticals, Inc. is considered one of the predominant lipid-based methods extensively employed for delivering nucleic acids systemically in various clinical trials. This technique encapsulates nucleic acids with high efficiency that are effective in delivering gene therapeutics especially to hepatocytes. TKM-PLK1 specifically targets polo-like kinase 1 (PLK1), a protein critical in driving tumor cell proliferation and recognized as a validated target within oncology. Tekmira has begun a phase I/II clinical trial testing TKM-PLK1 in patients diagnosed with gastrointestinal neuroendocrine tumors, adrenocortical carcinoma and hepatocellular carcinoma [142]. A summary of various LNPs in clinical trials can be found in Table 2.

## 7. USFDA-Approved LNPs for Biologics

The existing USFDA-approved LNP formulations consist of lipids which comprise (i) an ionizable cationic lipid; (ii) helper lipids, such as 1,2-distearoyl-sn-glycero-3-phosphocholine (DSPC); (iii) cholesterol, which also serves as a helper lipid; and (iv) a polyethylene glycol (PEG)–lipid conjugate [151].

### 7.1. Role of Ionizable Cationic Lipids

Ionizable lipids have a significant role in nucleic acid encapsulation. The ionizable lipid used in Comirnaty^®^, Spikevax^®^ and in Onpattro^®^ are ALC-0315, SM-102 and MC3. In general, cationic lipids typically consist of quaternary ammonium groups that have been alkylated, resulting in lipids with the same charge across different pH levels. In contrast, at a low pH (<6.0), ionizable lipids exhibit a positive charge due to the protonation of the free amine in the acidic condition. The charge of the ionizable lipids decreases or tends towards neutrality when the pH reverts to a physiological state (approximately pH 7.4). The ability of ionizable lipids to change their charge provides significant benefits in mRNA delivery and also as a key element of the endosomal escape of LNPs. Primarily, altering the lipid charge at different pH levels allows for a high efficiency in encapsulating mRNA. This is because the positively charged lipid can effectively interact with the negatively charged mRNA at low pH levels [152,153]. Ionizable lipids, which all contain esters, possess the ability to undergo hydrolysis degradation. Notably, even though MC3 is not fully biodegradable, and its metabolites endure in rats and non-human primates, the doses necessary for clinical use do not exhibit any signs of toxicity [154]. Following intravenous and intramuscular administration, biodegradable ionizable lipids incorporating ester bonds have demonstrated rapid elimination and excretion, alongside substantial tolerability, in both rodents and non-human primates [155].

### 7.2. Role of PEG–Lipids

PEG–lipids have a significant impact on the key characteristics of LNPs, which include particle size and polydispersity [156], the stability of LNPs during preparation and in storage, the prevention of aggregation [157], the encapsulation efficiency of nucleic acids, circulation half-life, immune response, in vivo distribution and efficiency in transfection [158,159]. The PEG–lipids used in Comirnaty^®^, Spikevax^®^ and in Onpattro^®^ are ALC-0159, PEG-DMG and PEG-c-DMG, respectively. The biological activity of LNPs is impacted by the structure of the lipid tail in PEG–lipids. Incorporation of PEG–lipids into the LNP membrane occurs through the hydrophobic tail (composed of alkyl/acyl chains). PEG–lipids possessing longer tails exhibit reduced propensity for dissociation from the LNP. Mui et al. demonstrated that PEG–lipid desorption from LNPs in circulation, analyzed one hour after administering them in vivo, was 45% for PEG–lipids containing C14 dialkyl chains, and conversely PEG–lipids with C16 and C18 dialkyl chains displayed only 1.3% and 0.2% desorption, respectively [160]. The short-chained diacyl PEG–lipid PEG-carbamate-1,2-dimyristoyl-sn-glycerol (PEG-c-DMG), which was introduced in the Onpattro^®^ formulation, tends to desorb from the LNPs upon intravenous administration. PEG–lipids with short lipid tails enable them to desorb from LNPs, allowing the particles to adsorb apoE, facilitating LNPs to endogenously target hepatocytes in the liver [32]. PEG–lipids serve to stabilize LNPs during their preparation and storage by creating a steric barrier that facilitates self-assembly and inhibits aggregation. Moreover, the specific type of PEG–lipid used has an influence on the circulation time of LNPs and their interactions with cells. The selection of a suitable PEG–lipid relies heavily on the therapeutic objective, the specific organ or cell type being targeted, and the method of administration. It is important to consider both the molar ratios and the length of the akyl/acyl chains that make up the lipid tail, as these factors affect important characteristics of lipid nanoparticles (LNPs).

### 7.3. Role of Helper Lipids

Helper lipids describe a range of lipids, such as sterols, phospholipids, and glycerolipids, which are typically characterized as non-cationic. Cholesterol, being an exchangeable molecule, has a significant effect on reducing the amount of protein that binds to the surface of liposome, thereby improving the circulation half-life. Hence, LNP formulations contain cholesterol in equimolar quantities compared to endogenous membranes. This serves to prevent any net efflux or influx and effectively preserves the integrity of the membrane. Thus, the incorporation of cholesterol into an LNP formulation serves to enhance its particle stability by actively regulating the integrity and rigidity of the lipid membrane. In study [9], it was highlighted that a certain threshold level of helper lipid is necessary to ensure stable encapsulation. More specifically, achieving near-complete siRNA encapsulation required a minimum cholesterol concentration of 40 mol% in the absence of any phospholipid. LNP formulations are incapable of maintaining a significant amount of cholesterol in a soluble state. The presence of a significantly greater molar quantity of cholesterol than that which can be stably maintained within a membrane is likely to lead to the formation of insoluble cholesterol crystallites in the core of LNPs, along with deprotonated ionizable lipid [161]. Moreover, the lipid’s elevated melting temperature (Tm) imparts a higher level of stability to the structure of the LNPs. Endosomal escape is hindered as a result of the high stability, which prevents the fusion of membranes with the endosomal membrane [162]. DOPE is a fusogenic lipid that consists of two unsaturated acyl chains containing a hydrocarbon chain with a single bond. Additionally, it has a relatively smaller head group, resulting in a cone-shaped geometry. DOPE undergoes a transition into a non-lamellar lipid phase due to the inverted hexagonal (HII) phase. This phase allows for membrane fusion and bilayer disruption, which leads to endosomal escape. Moreover, the presence of DOPE in cationic lipid formulations enhances transfection efficacy by aiding in membrane fusion [163,164]. Table 3 briefly summarizes the US FDA-approved LNP products.

BNT162b2 is a vaccine with nucleoside modifications carrying the genetic information for the spike glycoprotein of SARS-CoV-2, encapsulated within lipid nanoparticles. Following intramuscular injection, lipid nanoparticles act as a protective barrier against mRNA degradation, facilitating the effective delivery of mRNA into host cells. After a two-dose vaccination, each containing 30 µg per dose and administered 21 days apart, BNT162b2 demonstrated a 95% efficacy against COVID-19. The mRNA-1273 vaccine, to be precise, encodes the S-2P antigen, a vital element of the SARS-CoV-2 glycoprotein, along with a transmembrane anchor and the S1–S2 cleavage site encapsulated in the lipid nanoparticles. The mRNA-1273 vaccine, like the BNT162b2 vaccine, showed an incidence rate of adverse effects at around 12.6 cases per million doses (second dose) that were less severe and fatal, requiring no medical intervention [160,165,166].

## 8. Conclusions and Future Perspectives

Lipid-based drug delivery systems present an extensive array of formulation possibilities by potentially improving the bioavailability of biologics which arise from drug discovery process, which remains a significant challenge. The clinical success of mRNA-based vaccines for COVID-19 has propelled advancements beyond the state of the art in RNA-based therapies and has directed significant attention towards addressing the pressing and unmet requirements for advanced therapies. The expedited development and regulatory endorsement of mRNA vaccines have instilled confidence not only within the industry but also among regulators and the general public. At present, there is a unique opportunity to develop and extend the utilization of the LNP technology platform to address the challenges in the delivery of biologics. The lipophilicity of the drug candidate itself plays a crucial role in the potential effectiveness of lipid-based formulations to improve drug bioavailability. However, continued research is necessary to establish a more comprehensive and integrated approach for optimizing biologic formulations considering various critical parameters such as the selection of lipid excipients, physicochemical properties, functionalization, toxicity, drug loading capacity, drug dissolution profile, and intestinal stability for enhanced therapeutic outcomes. The USFDA has undertaken specific initiatives to establish guidelines for these LNPs, with the ultimate objective of making the regulatory filing process simpler and more streamlined. These guidelines provide manufacturers with an overview of the approach required to develop LNPs. They include specific tests and the exhaustive characterization involved in the biologic’s product development process, as well as methods to address existing challenges and concerns. However, there are several limitations associated with this technology, including challenges related to the stability of lipid-based formulations and manufacturing processes. These factors emphasize the necessity of addressing the development of appropriate regulatory guidelines for lipid-based formulations specifically for biologics to further advance the technology. To summarize, nano delivery systems of biologics are at advanced stage in the present scenario, corroborated by the promising results which mankind has observed recently during the COVID-19 pandemic, and with this success, LNPs potentially can be used for the delivery of many promising biological drugs. The advancement in the field is also dependent on various key factors such as the introduction and regulatory approval of novel smart lipids and the LNPs thereof. In future, advancements in the development of LNPs for the delivery of biological actives could help to treat the infectious, neglected, genetic, rare/orphan diseases and disorders which are difficult to treat or are untreatable presently.

The views expressed in the review have not been endorsed by a current or former employer and do not construe the employer’s position. The views are based on the author’s personal opinion and publicly available information.

## Figures and Tables

**Figure 1 pharmaceutics-16-00131-f001:**
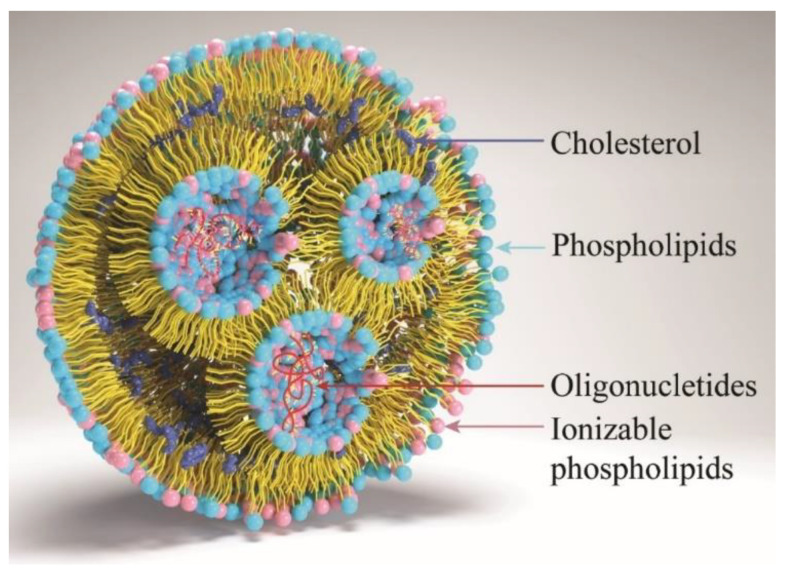
LNP structural components.

**Figure 2 pharmaceutics-16-00131-f002:**
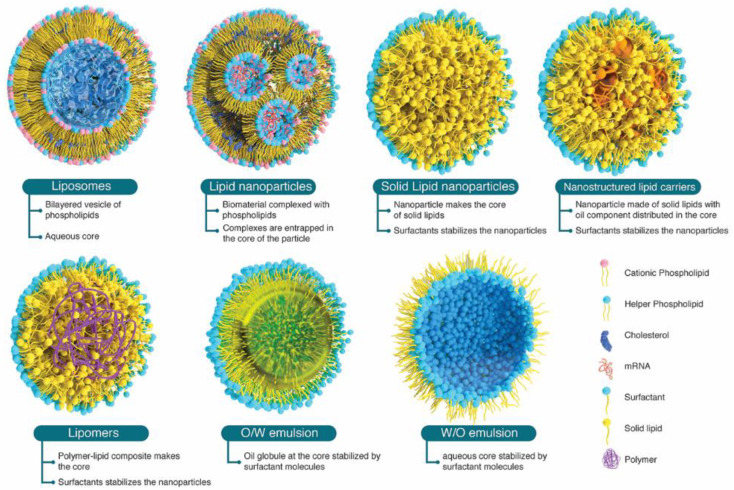
Different types of LNPs with characteristics.

**Figure 3 pharmaceutics-16-00131-f003:**
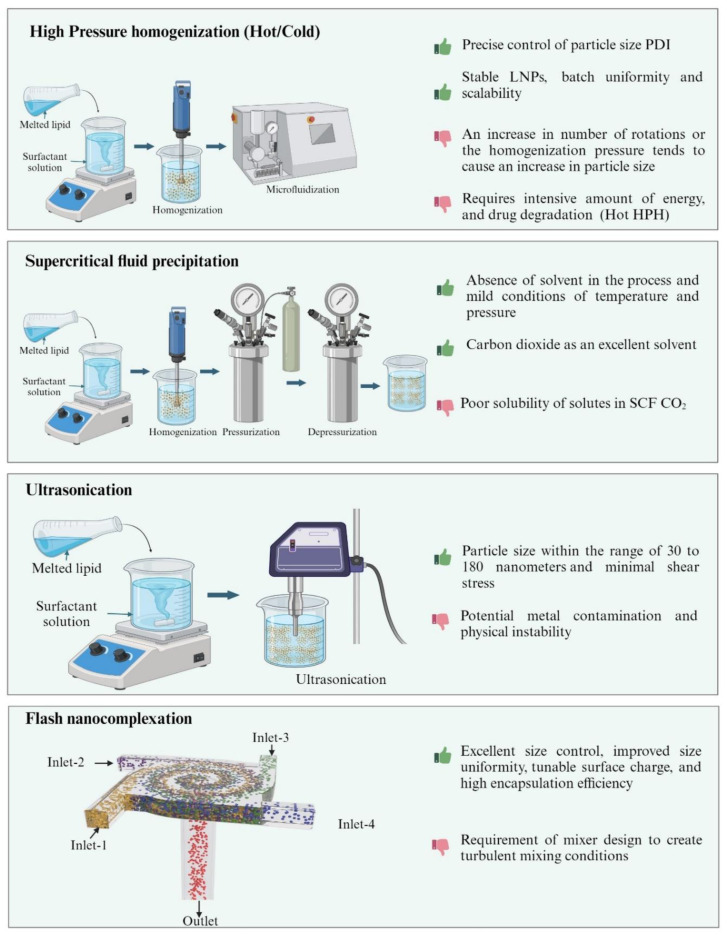
High-energy methods (Created with BioRender.com, accessed on 28 December 2023).

**Figure 4 pharmaceutics-16-00131-f004:**
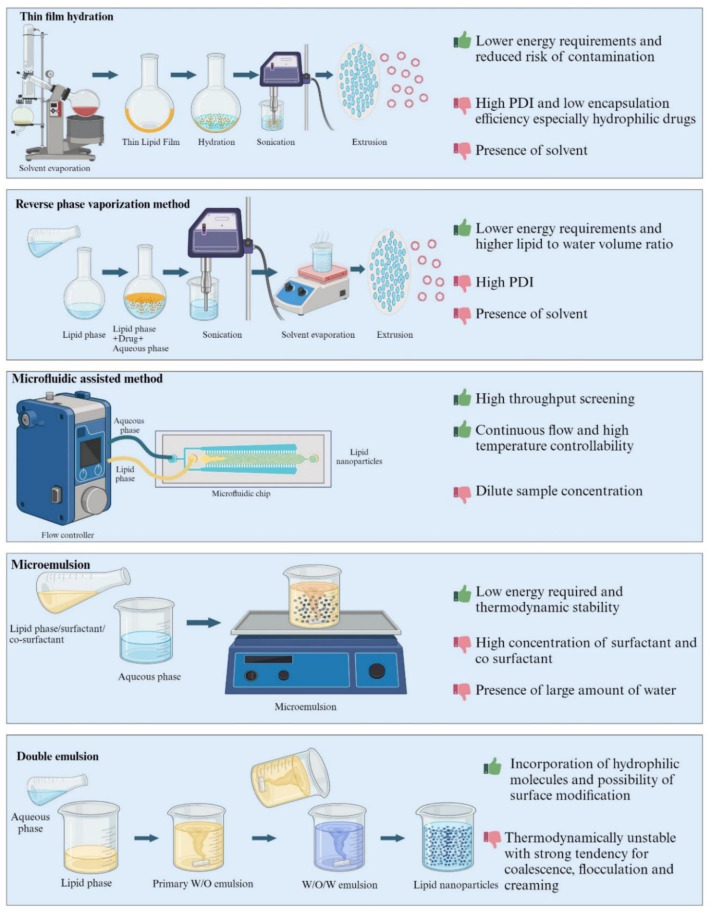
Low-energy methods (Created with BioRender.com, accessed on 28 December 2023).

**Figure 5 pharmaceutics-16-00131-f005:**
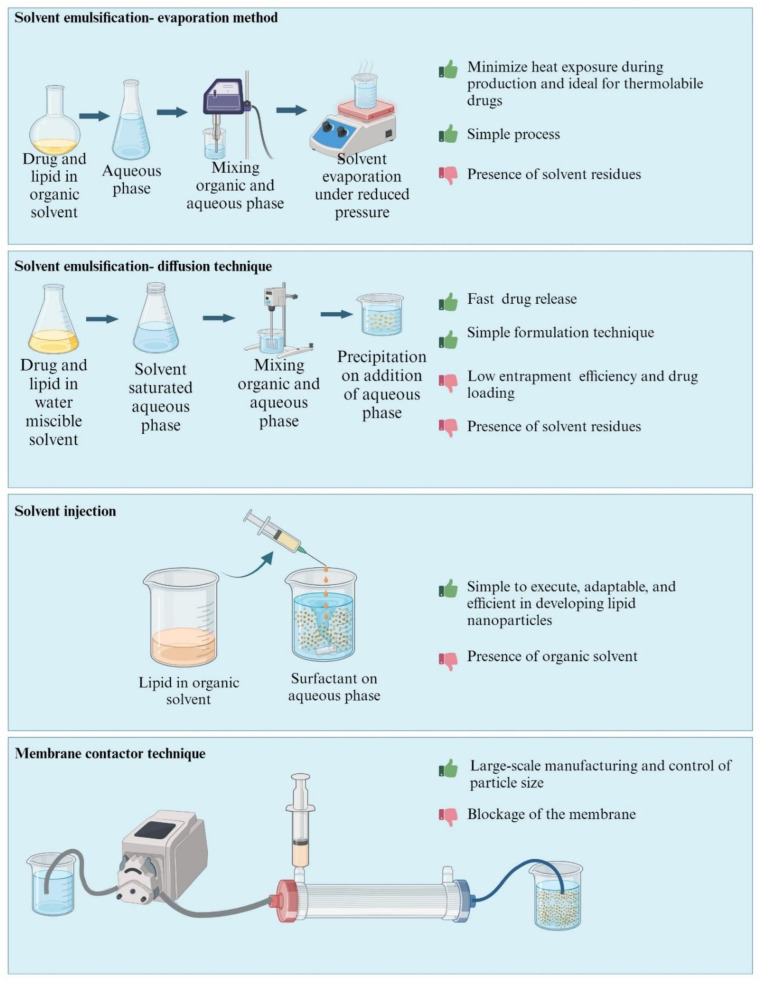
Organic solvent-based methods (Created with BioRender.com, (accessed on 28 December 2023).

**Table 1 pharmaceutics-16-00131-t001:** Lipids used in the development of LNP formulations.

Lipid Structure	Chemical Name	Structural Moieties/Characteristics	Main Findings	Ref.
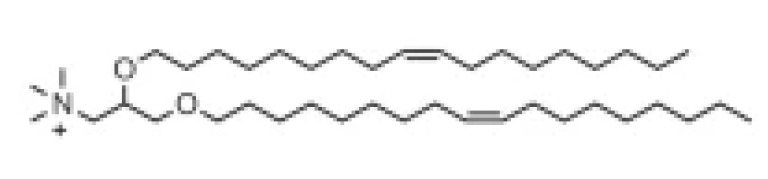	N-[1-(2,3-dioleyloxy) propyl]-N,N,N-trimethylammonium chloride(DOTMA)	Quaternary ammonium head groupGlycerol-based back boneTwo ether linkage bondsTwo hydrocarbon chains	DOTMA interacts directly with plasmid DNA, forming lipid–DNA complexes that exhibit 100% entrapment	[121]
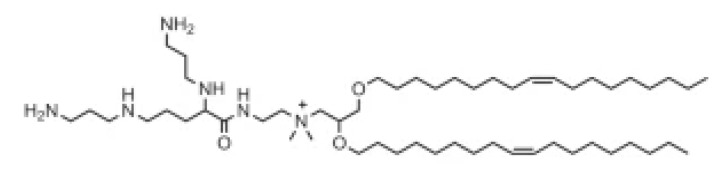	2,3-dioleyloxy-N-[2-(sperminecarboxamido) ethyl]-N,N-dimethyl-1-propanaminium trifluoroacetate(DOSPA)	Quaternary ammoniumSpermine head group bound via a peptide bond to the hydrophobic chainsTwo 18-carbon alkyl chains	Multivalent lipid, containing quaternary ammonium and spermine optimized to deliver mRNA in alveolar cells, cardiac muscle cells and pluripotent stem cells	[122]
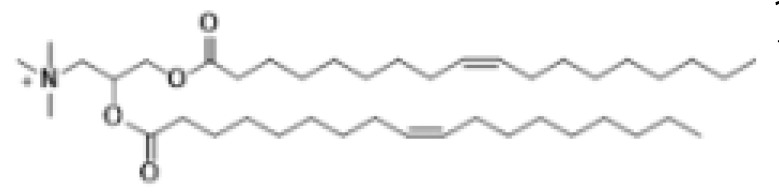	1,2-dioleoyl-3-trimethylammonium-propane(DOTAP)	Quaternary ammonium head groupGlycerol-based back boneTwo ester linkage bondsTwo hydrocarbon chains	Incorporates degradable ester bonds connecting the cationic head group and hydrophobic lipids exhibited elevated transfection efficiency and reduced toxicity	[123]
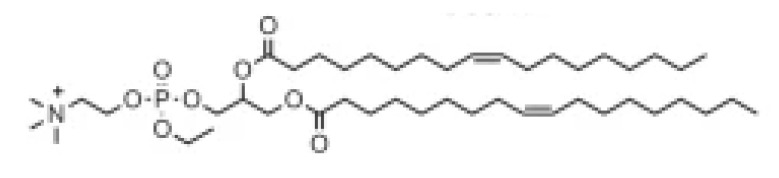	Ethyl phosphatidylcholine (ePC)	Quaternary nitrogen head group	Synthesized by introducing a third alkyloxy group into phosphatidylcholines to eliminate negative charge and applied for mRNA-based cancer immunotherapy and protein replacement therapies	[124,125]
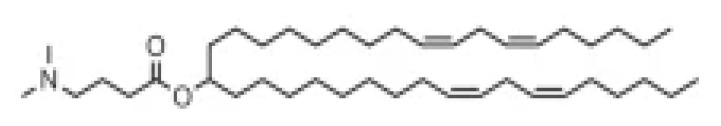	(6Z,9Z,28Z,31Z)-heptatriaconta-6,9,28,31-tetraen-19-yl 4(dimethyl amino) butanoate(DLin-MC3-DMA)	Tertiary amine head group which shows pH-dependent ionizationEster linkersTwo linoleyl tailspKa: 6.44	Employed in the liver’s mRNA delivery process through albumin receptor-mediated mechanisms.	[126]
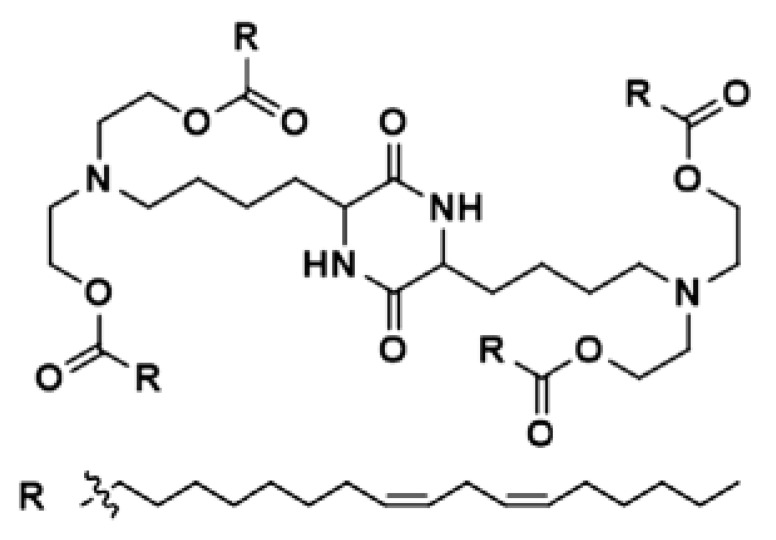	(((3,6-dioxopiperazine-2,5-diyl) bis (butane-4,1-diyl)) bis(azanetriyl)) tetrakis (ethane-2,1-diyl) (9Z,9′Z,9″Z,9‴Z,12Z,12′Z,12″Z,12‴Z)-tetrakis (octadeca-9,12-dienoate)(OF-Deg-Lin)	Diketopiperazine coreAlkenyl amino alcohols and containing four degradable ester linkagesDoubly unsaturated tailspKa: 5.7	Achieving specific and targeted delivery of mRNA into B lymphocytes.	[127]
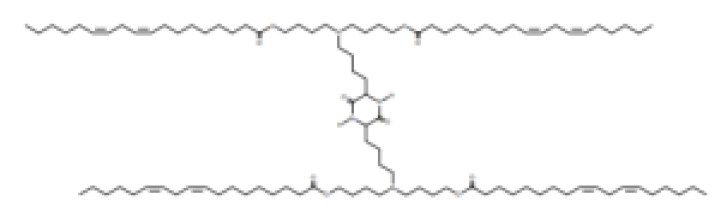	(((3,6-dioxopiperazine-2,5-diyl) bis(butane-4,1-diyl)) bis (azanetriyl)) tetrakis(butane-4,1-diyl) (9Z,9′Z,9″Z,9‴Z,12Z,12′Z,12″Z,12‴Z)-tetrakis(octadeca-9,12-dienoate)(OF-C4-Deg-Lin)	Diketopiperazine core containing doubly unsaturated tailLinkers containing a length of four-carbon aliphatic chain	Precision delivery of siRNAs and mRNAs.	[128]
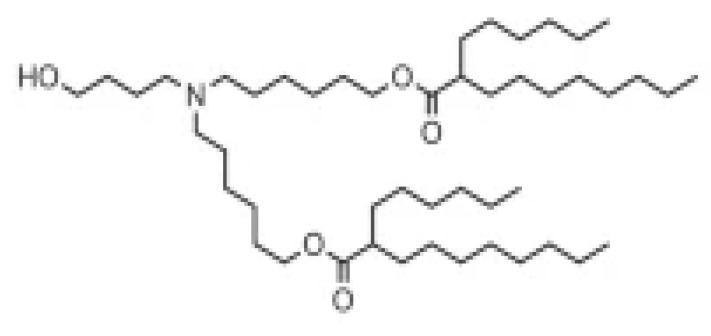	((4-hydroxy butyl) azanediyl) bis(hexane-6,1-diyl) bis(2-hexyldecanoate)(ALC-0315)	Tertiary amine head groupTerminal hydroxyl groupEster linkerspKa: 6.09	Ionizable delivery components in the mRNA-1273 and BNT162b COVID-19 vaccines.Nucleic acid complexation and Membrane fusion	[129]
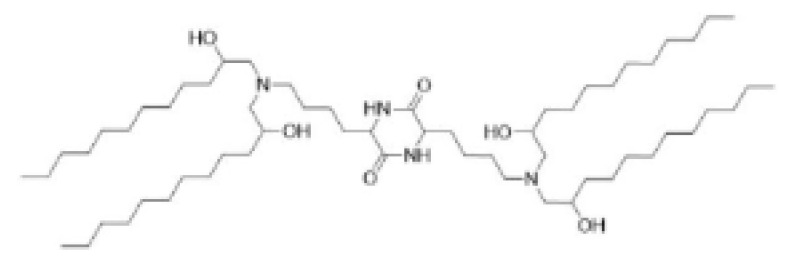	3,6-bis(4-(bis(2-hydroxydodecyl) amino) butyl) piperazine-2,5-dione(cKK-E12)	Diketopiperazine core-based headFour lipid tails	Multi tail ionizable lipid with enhanced endosome disrupting ability and effectively identified to target hepatic genes	[130]
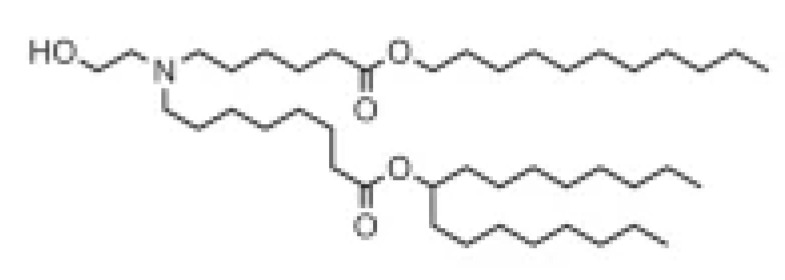	Heptadecan-9-yl 8-((2-hydroxyethyl) (6-oxo-6-(undecyloxy) hexyl) amino) octanoate(Lipid H (SM-102)	Tertiary amine head groupTerminal hydroxyl groupEster linkersTwo branched saturated tailspKa: 6.75	SM-102 contains a primary degradable ester tail and adegradable branched ester tail positively charged at low pH.	[131]
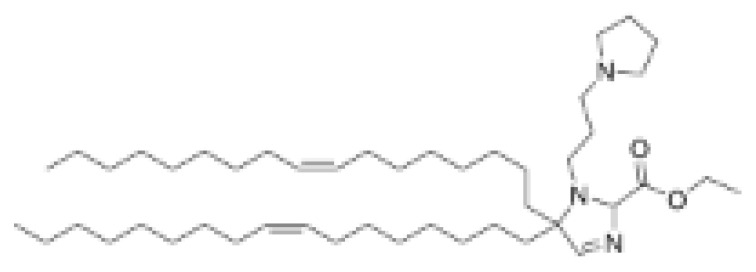	Ethyl 5,5-di((Z)-heptadec-8-en-1-yl)-1-(3-(pyrrolidin-1-yl) propyl)-2,5-dihydro-1H-imidazole-2-carboxylate(A2-Iso5-2DC18)	Cyclic amine head groupDihydroimidazole linkerUnsaturated lipid tail	Efficiently deliver mRNA and yield effective mRNA expression	[132]
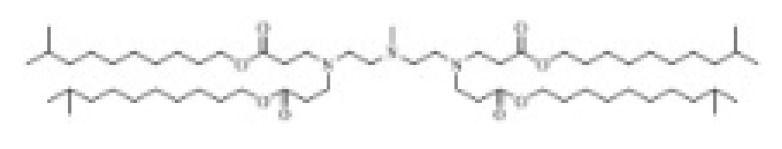	Tetrakis (8-methylnonyl) 3,3′,3″,3‴-(((methylazanediyl) bis (propane-3,1 diyl)) bis (azanetriyl)) tetrapropionate(306Oi10)	Branched tailIonizable lipid tail isodecyl acrylate	Efficiently co-delivered multiple RNA constructs achieving a transfection rate of over 80% in hepatocytes, Kupffer cells, and endothelial cells	[130]
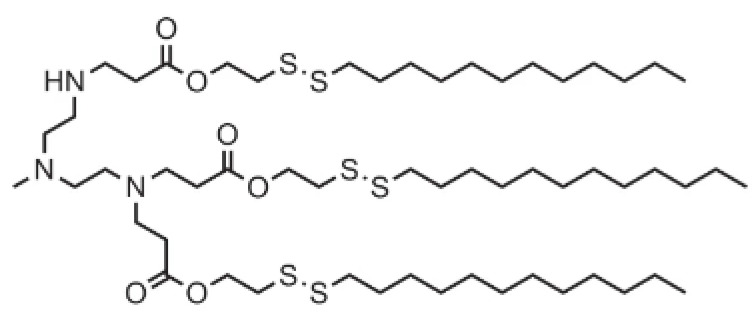	Bis (2-(dodecyl disulfanyl) ethyl) 3,3′-((3-methyl-9-oxo-10-oxa-13,14-dithia-3,6-diazahexacosyl) azanediyl) dipropionate(BAMEA-O16B)	Biodegradable lipid integrated with disulfide bonds containing hydrophobic tails	BAMEAO16B efficiently delivers Cas9 mRNA and sgRNA into cells while releasing RNA in response to the reductive intracellular environment for genome editing	[133]
, 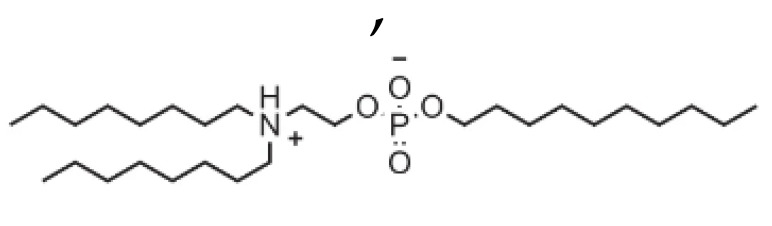	Decyl (2-(dioctylammonio)ethyl) phosphate(9A1P9)	Multi tail ionizable cationic phospholipidZwitterionic head and three tails	Assists to promote membrane destabilization and drug release	[130]
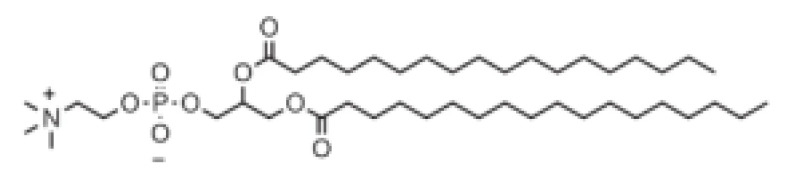	1,2-distearoyl-sn-glycero-3-phosphocholine(DSPC)	Quaternized amine phosphatidylcholine head groupSaturated acyl chains	Phosphatidylcholine with cylindrical geometry that allows DSPC molecules to form a lamellar phase, which stabilizes the structure of lipid nanoparticles	[99]
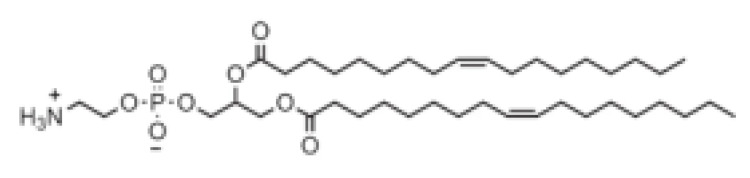	1,2-dioleoyl-sn-glycero-3-phosphoethanolamine(DOPE)	Primary amine phosphatidylethanolamine head groupDouble-unsaturated lipid tail	Improves the transfection efficiency in cationic lipid formulations by promoting membrane fusion.	[134]
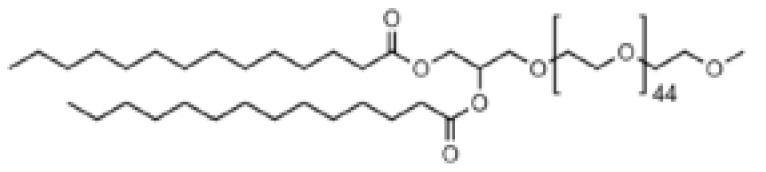	1,2-dimyristoyl-rac-glycero-3 methoxy polyethylene glycol-2000 (PEG2000-DMG)	PEGylation of myristoyl diglyceride	PEG modifications increase the nanoparticles’ blood circulation duration by diminishing clearance through the kidneys and the mononuclear phagocyte system.	[130]
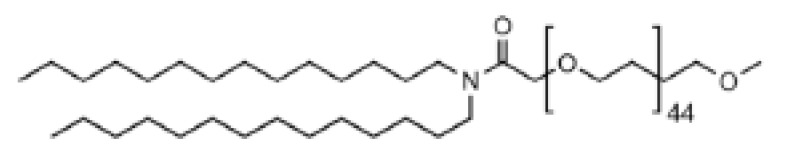	2-[(polyethyleneglycol)-2000]-N, N ditetradecyl acetamide(ALC-0159)	PEGylated lipid consisting of PEG group conjugated to a lipid anchor with two 14-carbon saturated alkyl chains
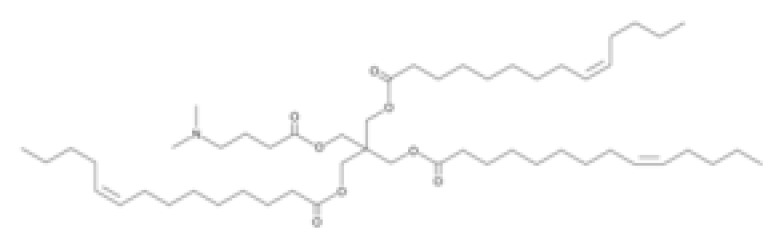	2-(((4 (dimethylamino) butanoyl)oxy)methyl)-2-((((Z)-tetradec-9-enoyl)oxy)methyl) propane-1,3-diyl (9Z,9′Z)-*bis*(tetradec-9-enoate)(TCL053)	Three-tailed ionizable lipidpKa: 6.8	TCL053 iLNPs transiently deliver CRISPR-Cas9mRNA and sgRNA to multiple muscle tissues, reducingimmunogenicity and increasing the safety of iLNPs.	[135]
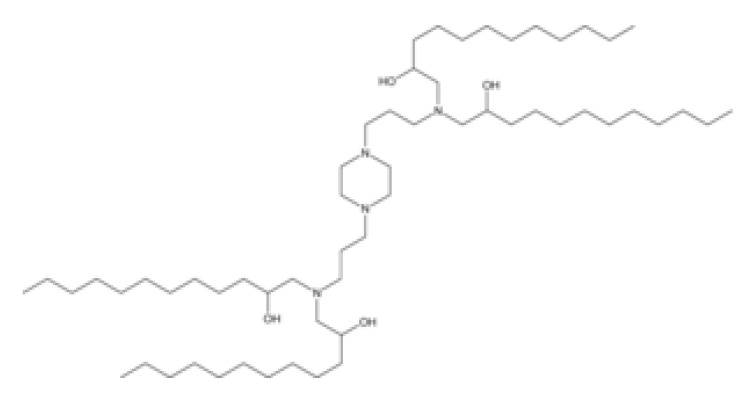	1,1′,1′′,1′′′-[1,4-piperazinediylbis(3,1-propanediylnitrilo)] tetrakis-2-dodecanol(246C10)	Lipid based on piperidinepKa: 6.75	Buffer preparation of 246C10 iLNPs could increase the encapsulation efficiency of CRISPR-Cas9 mRNA and sgRNA. These iLNPs were able to treat hemophilia safely, without causing hepatotoxicity.	[136]
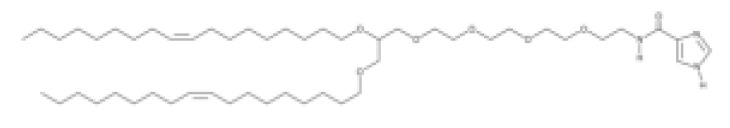	N-[(25Z)-14-[(9Z)-9-octadecen-1-yloxy]-3,6,9,12,16-pentaoxatetratriacont-25-en-1-yl]-1H-imidazole-5-carboxamide(DOG-IM4)	Imidazole head and dioleoyl lipid tailShort flexible polyoxyethylene linkerpKa: 5.6	Ionizable cationic lipid for the formulation of LNPs displaying increased stability at 4 °C in liquid form and promoting robust mRNA expression and strong immune responses	[137]

**Table 2 pharmaceutics-16-00131-t002:** LNPs in clinical trial/phase.

Trade Name	Drug/Target	Clinical Trial	Indication	Reference
TKM-080301 Arbutus Biopharma Corporation, Warminster Township, PA, USA	PLK1 (polo-like kinase-1)	Phase I/II	Gastrointestinal neuroendocrine tumors, adrenocortical carcinoma tumors, advanced hepatocellular carcinoma	[143] ClinicalTrials.gov Identifier: NCT01262235, NCT02191878
EphA2 siRNA M.D. Anderson Cancer Center National Cancer Institute (NCI), Bethesda, MD, USA	EphA2	Phase I	Advanced Malignant Solid Neoplasm	[144] ClinicalTrials.gov Identifier: NCT01591356
ARB-001467 Arbutus Biopharma Corporation, Warminster Township, PA, USA	HBsAg	Phase II	Hepatitis B, Chronic	[145] ClinicalTrials.gov Identifier: NCT02631096
ALN-PCS02 Alnylam Pharmaceutical, Cambridge, MA, USA	PCSK9	Phase I	Elevated LDL cholesterol	ClinicalTrials.gov Identifier: NCT01437059
ALN-VSP02 Lipid Nanoparticle Alnylam Pharmaceutical, Cambridge, MA, USA	siRNA-KSP	Phase I	Cancer—Solid tumors	[146] ClinicalTrials.gov Identifier: NCT01158079
Liposomal T4N5 Lotion National Cancer Institute (NCI), Bethesda, MD, USA	a prokaryotic DNA repair enzyme	Phase II	The Recurrence of Nonmelanoma Skin Cancer	[147] ClinicalTrials.gov Identifier: NCT00089180
Lipo-MERIT BioNTech RNA Pharmaceuticals GmbH Mainz, Germany	ribonucleic acid (RNA)-drug products (DPs)	Phase I	Cancer (melanoma)	ClinicalTrials.gov Identifier: NCT02410733
mRNA-4157 Moderna TX, Inc. and Merck Sharp & Dohme Corp, Cambridge, MA, USA	mRNA	Phase II	Cancer (bladder carcinoma, melanoma and non-small-cell lung carcinoma (NSCLC))	ClinicalTrials.gov Identifier: NCT03313778 NCT03897881
Liposome Encapsulated Mitoxantrone INSYS Therapeutics Inc., Chandler, AR, USA	Mitoxantrone	Phase I	Solid Tumors	[148] ClinicalTrials.gov Identifier: NCT00024492
S-CKD602 University of Pittsburgh, Pittsburgh, PA, USA ALZA	CKD-602	Phase I	Advanced Malignancies	[149] ClinicalTrials.gov Identifier: NCT00177281
Topotecan Liposomes Injection (TLI) Spectrum Pharmaceuticals, Inc., Boston, MA, USA	Topotecan	Phase I	Small Cell Lung Cancer Ovarian Cancer Solid Tumors	ClinicalTrials.gov Identifier: NCT00765973
TLD-1 Swiss Group for Clinical Cancer Research, Effingerstrasse 33 CH-3008 Bern, Switzerland	Doxorubicin	Phase I	Advanced Solid Tumors	ClinicalTrials.gov Identifier: NCT03387917
LEP-ETU INSYS Therapeutics Inc., Chandler, AR, USA	Paclitaxel	Phase I/II	Advanced cancer (Neoplasm) Metastatic Breast Cancer	ClinicalTrials.gov Identifier: NCT00080418 NCT01190982 NCT00100139
MBP-426^®^ Mebiopharm Co., Ltd. Tokyo, Japan	Oxaliplatin/Transferrin	Phase I/II	Solid Tumors	ClinicalTrials.gov Identifier: NCT00964080
OSI-211 Astellas Pharma Inc., OSI Pharmaceuticals, Northbrook, IL, USA	Lurtotecan	Phase II	Recurrent Small Cell Lung Cancer	ClinicalTrials.gov Identifier: NCT00046787
ThermoDox^®^ Celsion, Lawrenceville, NJ, USA	Doxorubicin/Targeted thermal therapy	Phase III	Hepatocellular Carcinoma	ClinicalTrials.gov Identifier: NCT02112656
SPI-77 NYU Langone Health and National Cancer Institute (USA), Betshsda, MD, USA	Cisplatin	Phase II	Ovarian Cancer	ClinicalTrials.gov Identifier: NCT00004083
LiPlaCis Oncology Venture/ Allarity Therapeutics, Boston, MA, USA	Cisplatin	Phase I/II	Advanced or Refractory Solid Tumors/Metastatic Breast Cancer, Prostate Cancer and Skin Cancer	ClinicalTrials.gov Identifier: NCT01861496
BNT162a1 BNT162b1 BNT162c2 BioNTech SE, Mainz, Germany	nucleoside-modified mRNA	Phase I/II Phase II/III	SARS-CoV-2	[150] ClinicalTrials.gov Identifier: NCT04380701 NCT04368728
ARCT-021 (mRNA Lunar-Cov19) Arcturus Therapeutics, San Diego, CA and Duke-NUS, USA	mRNA	Phase I/II	SARS-CoV-2	ClinicalTrials.gov Identifier: NCT04480957
ALN-TTR02; Alnylam, Cambridge, MA, USA	Transthyretin	Phase II/III	Transthyretin-mediated amyloidosis	ClinicalTrials.gov Identifier: NCT01617967 NCT01960348
TKM-100201; Tekmira/Arbutus Biopharma Corporation, Warminster Township, PA, USA	VP24, VP35, Zaire Ebola L-polymerase	Phase I	Ebola virus infection	ClinicalTrials.gov Identifier: NCT01518881
PRO-040201; Tekmira/Arbutus Biopharma Corporation, Warminster Township, PAv	ApoB	Phase I	Hypercholesterolemia	ClinicalTrials.gov Identifier: NCT00927459
TKM 080301; Tekmira/Arbutus Biopharma Corporation, Warminster Township, PA, USA	Polo-like kinase 1	Phase I/II	Neuroendocrine tumors; adrenocortical carcinoma	ClinicalTrials.gov Identifier: NCT01262235 NCT01437007
siRNA-EphA2-DOPC; M.D. Anderson Cancer Center	Ephrin type-A receptor 2	Phase I	Advanced cancers	ClinicalTrials.gov Identifier: NCT01591356
Atu027; Silence Therapeutics, London, UK	Protein kinase N3	Phase I	Advanced solid tumors	ClinicalTrials.gov Identifier: NCT00938574
NTLA-2001 Intellia Therapeutics, Cambridge, MA, USA	Cas9 mRNA	Phase I	Amyloidosis	ClinicalTrials.gov Identifier: NCT04601051
mRNA-2752, ModernaTX, Inc.; Cambridge, MA, USA	Human OX40L, IL-23, and IL-36γ	Phase I	Solid Tumor	ClinicalTrials.gov Identifier: NCT03739931
INT-1B3 InteRNA, Utrecht, The Netherlands	miR-193a-3p	Phase I	Solid Tumor	ClinicalTrials.gov Identifier: NCT04675996
Acclaim-1 Genprex, Inc. Austin, TX, USA	TUSC2/ DNA plasmid	Phase I/II	Carcinoma, Non-Small Cell Lung	ClinicalTrials.gov Identifier: NCT04486833
mRNA-1944 Moderna TX, Inc. Cambridge, MA, USA	mRNA encoding for CHKV-24 immunoglobulin G	Phase I	Chikungunya virus	ClinicalTrials.gov Identifier: NCT03829384
mRNA-3745 Moderna TX, Inc. Cambridge, MA, USA	mRNA encoding humanG6Pase-α S298C	Phase I	Glycogen storage disease	ClinicalTrials.gov Identifier: NCT 05095727
mRNA-1647 Moderna TX, Inc. Cambridge, MA, USA	mRNA Cytomegalovirus	Phase III	Cytomegalovirus infection	ClinicalTrials.gov Identifier: NCT05085366
mRNA-1893 Moderna TX, Inc. Cambridge, MA, USA	mRNA ZIKV vaccine	Phase II	Zika virus	ClinicalTrials.gov Identifier: NCT04917861

**Table 3 pharmaceutics-16-00131-t003:** List of USFDA-approved LNPs for biologics.

Trade Name	Payload	Excipients	Approved Indication	Route of Administration	BLA Number	Approval Year	Applicant Holder/Reference
Comirnaty	Tozinameran (mRNA)	(i) ALC-0315 = (4-hydroxybutyl) azanediyl) bis (hexane-6,1-diyl) bis(2-hexyldecanoate), (ii) ALC-0159 = 2-[(polyethylene glycol)-2000]-N, N di tetradecyl acetamide, (iii) 1,2-Distearoyl-*sn*-glycero-3-phosphocholine (DSPC), (iv) Cholesterol	COVID-19 immunization	Intramuscular (i.m)	125,742	2021	BioNTech Mainz, Germany https://www.fda.gov/vaccines-blood-biologics/comirnaty (accessed on 15 January 2024)
Onpattro	Patisiran sodium (siRNA)	(i) DLin-MC3-DMA: (6Z,9Z,28Z,31Z)-heptatriaconta-6,9,28,31-tetraen-19-yl-4-(dimethylamino) butanoate, (ii) 1,2-Distearoyl-sn-glycero-3-phosphocholine (DSPC), (iii) PEG2000-DMG: Alpha-(3′-{[1,2-di(myristyloxy) propanoxy] carbonyl amino} propyl)-ω-methoxy polyoxyethylene Cholesterol	Transthyretin (TTR)-mediated amyloidosis	Intravenous (iv) infusion	N210922	2018	Alnylam Pharmaceuticals, Inc. Cambridge, MA FDA approves first-of-its kind targeted RNA-based therapy to treat a rare disease | FDA)
Spikevax	Elasomeran/ (mRNA-1273)	(i) SM-102 (heptadecan-9-yl 8-((2-hydroxyethyl) (6-oxo-6-(undecyloxy) hexyl) amino) octanoate}, (ii) PEG2000-DMG:1-monomethoxypolyethyleneglycol-2,3-dimyristylglycerol with polyethylene glycol of average molecular weight 2000 (iii) 1,2-Distearoyl-sn-glycero-3 phosphocholine (DSPC) (iv) Cholesterol	COVID-19 immunization	Intramuscular (i.m)	125,752	2022	ModernaTX, Inc. https://www.fda.gov/vaccines-blood-biologics/spikevax (accessed on 15 January 2024)

## Data Availability

Data can be found within the article.

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
