# Peer review of "Chemistry and Art of Developing Lipid Nanoparticles for Biologics Delivery: Focus on Development and Scale-Up"

_pharmaceutics, 2024, doi:10.3390/pharmaceutics16010131_

Round 1

Reviewer 1 Report

Comments and Suggestions for Authors

This is a comprehensive survey of the various methods used for the manufacture of LNPs, with some critique of their suitability for scaling up from the lab to large scale manufacture. There is very little description of their use for delivery of biologics though (beyond a list of nucleic acid payloads in the table concerning ionizable lipids), even though they are named in the title. There is some additional mention of biologics (but not exclusively) in the rather thin section on LNP formulations in clinical trials and FDA approved medicines - again in the form of a table. Some additional comment would be useful in this section. The section on LNP chemistry is also quite sparse, perhaps the sections on the characterisitcs of the different lipidic components could be incorporated into the chemistry section.

Comments on the Quality of English Language

I suggest the review should be thoroughly proofread.

Author Response

Please see attached word document for response.

Reviewer 2 Report

Comments and Suggestions for Authors

This review collects and summarizes interesting information on methodology, materials and procedures used to produce lipid nanoparticles (LPNs) intended for drug delivery. In general terms, it is a useful piece of work that may serve the readers to easily get a general view on the different strategies, caveats and particularities that need to be taken into account when producing LPNs at different scales and contexts. The authors may still like to consider completing a few questions in order to enhance the relevance and utility of their review:

A list of the abbreviations used along the whole text would be the most useful for the readers to follow the content easily.

scCO2 (page 5, line 178) I guess means “super critical” CO2. Please, clarify.

The authors could considering including an additional Table summarizing basic concepts of LNPs produced by the different methods. For instance, joining the systematic information with respect to for instance typical LPN size, the feasibility of application at different scales, or the caveats, advantages and disadvantages of the different methods.

Please, provide magnitudes in units of the S.I.

Please, comment on the feasibility of filtration methods to get sterility of LPNs formulations also against the presence of viruses.

Please, consider discussion of the problems with thermal treatments, including lyophilization that requires freezing to very low temperatures, potentially affecting phase behavior and phase separation in lipid and lipid mixtures.

The authors may like adding a comment that structural characterization of LNPs using AFM also allows their analysis in different physiological-like environment such as aqueous solutions.

The mention to Drug release behavior (section 3.4) should perhaps distinguish between “spontaneous” time-dependent delivery, related with LNPs stability, and “desired” release triggered by any condition that looks for a progressive, or busted liberation of the encapsulated drug in defined therapeutic contexts.

A comment on the potential problems with lipid oxidation should be warranted within the discussion on LPN stability.

The discussion on the fate of delivered LPNs should also mention the clearance from blood exerted by spleen.

Although it has not been mentioned explicitly, the notion of the biological “corona” consisting of the proteins and other molecules adsorbing on the surface of NPs exposed to different biological media, is pertinent also for LNPs and their fate. The authors may like to include a explicit paragraph in that respect.

Page 21, line 720-722: “…DSPC phospholipid contains saturated acyl chains, with one or more double bonds, in the tail of the lipid and larger head group that forms a cylinder-shaped geometry”. If DSPC means distearoylPC, stearic acid chains are certainly saturated and contain no double bonds. Please, correct.

Author Response

(The authors gave the same response as above.)

Round 2

Reviewer 1 Report

Comments and Suggestions for Authors

No further modifications required.